



# Summer precipitation reconstructed quantitatively using a Mid Holocene δ¹³C common millet record from Guanzhong Basin, China

Qing Yang[1, 2], Xiaoqiang Li[2]*, Xinying Zhou[2], Keliang Zhao[2] & Nan Sun[3]

1. School of Geography Science, Nanjing Normal University, Nanjing, 210023, China
2. Key Laboratory of Vertebrate Evolution and Human Origin of Chinese Academy of Sciences,
   Institute of Vertebrate Paleontology and Paleoanthropology, Chinese Academy of Sciences,
   Beijing, 100044, China
3. The School of Earth Science and Resources, Chang'an University, Xi'an, Sha'anxi, 710054,
   China

*Correspondence and requests for materials should be addressed to X.L. (lixiaoqiang@ivpp.ac.cn)

## Abstract

In order to produce quantitative Holocene precipitation reconstructions for particular geographical areas, explicit proxies and accurate dating controls are required. The fossilized seeds of common millet (*Panicum miliaceum*) are found throughout the sedimentary strata of northern China, and are highly suited to the production of accurate quantitative Holocene precipitation reconstructions: their isotopic carbon composition ($\delta^{13}C$) gives a measure of the precipitation required during the growing season, and allows these seeds to be dated. We therefore used a robust regression function, as part of a systematic study of the $\delta^{13}C$ of common millet, to produce a quantitative reconstruction of Mid Holocene summer precipitation in the Guanzhong Basin. Our results showed that summer precipitation from 7.7-3.4 ka BP was 240-477 mm, with a mean of 354 mm, *i.e.* ～50 mm or 17% higher than present levels. Maximal mean summer precipitation peaked at 414 mm, ~109 mm (or 36%) higher than today, occurring during 6.4-5.5 ka BP; this is when the East Asian Summer Monsoon (EASM) was at its peak. As the $\delta^{13}C$-based precipitation record can reliably indicate EASM intensity during the Holocene, this work can provide a reliable proxy for further research into the detailed processes, and precise mechanisms, of the EASM.

**Keywords**: summer precipitation; quantitative reconstruction; Holocene; Chinese Loess Plateau; common millet; stable carbon isotope

**Sect.**

## 1 Introduction

The reconstruction of global climate changes through history is an important part of the Past Global Changes (PAGES) project. Such reconstructions provide an attempt to understand the evolution of the earth's climate over time, and any temporal, spatial and/or regional differences. Quantitatively reconstructing climatic factors such as temperature and precipitation provides an understanding of agricultural development and the human impact upon the landscape and environment. However, modern records are insufficient for the accurate documenting of the drivers of climate change, since they cover only the past century or so (De Menocal, 2001). The quantitative





reconstruction of temperature and precipitation using high resolution climatic proxy
records has therefore become the principal thrust of PAGES research. To this end, we
chose an area key to the warm period of the Holocene to produce a quantitative
precipitation reconstruction for that geological time.
The Holocene, as the most recent geological period, has witnessed the
emergence and rapid development of primitive farming, and therefore has the closest
relation to human survival and development. The Holocene has experienced three
distinct climatic phases: a rapid warming during the Early Holocene; a warm and
humid Mid Holocene phase; and a gradual cooling during the Late Holocene (An *et*
*al*., 2000; Wang *et al*., 2005a). The Holocene has also seen climatic fluctuations
caused by rapid climate change and other catastrophic events such as flooding,
drought and intense tropical cyclones (Bond *et al*., 1997; Mayewski *et al*., 2004;
Kleiven *et al*., 2008). The megathermal phase of the Mid Holocene is similar to a
scenario whereby global average temperatures rise by $1\sim2°C$ (Prentice and Webb,
1998), and can therefore be regarded as a paleoanalog for future climate prediction
and the evaluation of potential environmental impacts.
Centennial scale quantitative paleoclimatic reconstructions are needed if we are
to understand more completely and accurately climate change processes and controls.
To date, most continental paleoclimate studies have focused on temperature (Porter
and An, 1995; Guo *et al*., 1996; Genty *et al*., 2003; Wang *et al*., 2008; Sun *et al*.,
2012). Reconstructions of regional and global temperatures have shown that the warm
Early Holocene (10-5 ka BP) was followed by a 0.7°C cooling through the Mid to
Late Holocene (<5 ka BP), reaching a minimum ~200 years ago, during the 'Little Ice
Age'. The recent rapid warming of the climate is unprecedented, based on surface
temperature reconstructions for the past 1,500 years (Marcott *et al*., 2013). However,
the increase in global surface temperatures has tended to cause changes in
precipitation and atmospheric moisture through changes in atmospheric circulation, a
more active hydrological cycle and an increasing water holding capacity throughout
the atmosphere (Dore, 2005). The availability of water, one of the major challenges
for the future, cannot be ignored, due to its significant role in the hydrological cycle
(Hetté and Guiot, 2005).
The EASM, an important component of the Asian Summer Monsoon (ASM),
plays an indispensable role in the hydrological cycle over southern China. Various
proxies have been adopted in studies of the EASM during the Holocene. Due to their
reliable chronology and relatively easy dating, oxygen isotopes ($\delta^{18}O$) in speleothems
from Chinese caves have been taken as a robust measure of summer monsoon
precipitation values (Wang *et al*., 2005a; Cheng *et al*., 2009). However, the
interpretation of these changes in the $\delta^{18}O$ values of precipitation remains highly
controversial; some scientists have contended that the stalagmite $\delta^{18}O$ record from the
EASM region may not record EASM variability (Le Grande and Schmidt, 2009;
Maher and Thompson, 2012; Tan, 2012; Caley *et al*., 2014; Liu *et al*., 2015).
The Chinese Loess Plateau (CLP), located in a transition zone between a semi-
arid and semi-humid climate, and being deeply impacted by the EASM, is, and has
been, highly sensitive to changes in precipitation and has thus long been a key area for



EASM research. The quantitative precipitation reconstruction results obtained have
been based exclusively on climatic proxies derived from the CLP's geological and
biological records. In the western CLP, fossil charcoal records in the Tianshui Basin
have demonstrated that the mean annual precipitation (MAP) was 688-778 mm
between 5.2-4.3 ka BP (Sun and Li, 2012). In the CLP's hinterland, magnetic
susceptibility records from the Luochuan profile have provided estimates of Holocene
MAP varying between 600 and 750 mm, with a mean value of 701±74 mm (Lu $et$ $al$.,
1994). In the southern CLP, Guanzhong Basin MAP, as revealed by plant phytolith
assemblies, was 700–800 mm during the Holocene, indicating a much more humid
climate than today's (Lu $et$ $al$., 1996). Further evidence from the transfer functions of
geological records and the intensity of pedogenesis has shown that Guanzhong Basin
MAP was >700 mm during the Holocene Optimum (Sun $et$ $al$., 1999; Zhao, 2003),
supporting the aforementioned results. However, due to their intrinsic limitations,
such as discontinuity and an indefinite response mechanism between the proxies and
climate change, these tentative proxies have not been extensively applied. This has led
to either short-term or/and unverifiable quantitative results. Selecting an effective
proxy which evinces a continuous distribution, reliable dating and an unambiguous
implication is crucial for the quantitative reconstruction of paleoprecipitation. High-
resolution pollen-based quantitative precipitation results indicating EASM evolution
have recently been obtained from an alpine lake in northern China (Chen $et$ $al$., 2015).
However, because these are attributable solely to this unique environment, a regional
quantitative precipitation reconstruction, and therefore a new proxy, is still required.
Common millet ($Panicum$ $miliaceum$), as the most representative agricultural
rain-fed crop of northern China, contains $\delta^{13}C$; this is sensitive to precipitation and
can thus effectively record precipitation during the growing season (Yang and Li,
2015). Rain-fed agriculture originated in the CLP, giving rise to the first
recognizably Chinese civilization. Many archeological relics from an unbroken
historical continuum are therefore found throughout the region (An, 1988).
Quantities of the fossilized seeds of common millet are well-preserved in the cultural
layers of these archeological sites (Zhao and Xu, 2004; Liu $et$ $al$., 2008; Lu $et$ $al$.,
2009). Their stable $\delta^{13}C$ compositions, which remain little change because of the low
temperatures associated with carbonization, contain valuable information about
paleoclimate change and early agricultural activities (Yang $et$ $al$., 2011a, 2011b).
Common millet remains are therefore perfect for quantitatively reconstructing
Holocene precipitation in the CLP.
The Guanzhong Basin (Figure 1), in the southern CLP, was the cradle of
Neolithic culture and China's ancient civilization, and fostered the Laoguantai (~7.8-
6.9 ka BP), Yangshao (~6.9-5.0 ka BP) and Longshan (~5.0-4.0 ka BP) cultures (Ren
and Wu, 2010), the pre-Zhou culture (~3.5-3.0 ka BP) (Lei, 2010), and the Zhou
dynasties. Due to the intensity of early agricultural activity, huge quantities of
common millet remains have been preserved in numerous, continuously-occupied
cultural sites. Carbonized common millet seeds are the most abundant resource found
in the samples collected in this study from these cultural layers.
In this study, common millet remains, from five sections characterized by





continued and well-developed sedimentation at typical archeological sites, including the Baijiacun (BJC), Huiduipo (HDP), Manan (MN), Beiniu (BN) and Nansha (NS) sites (Figure 1), were sampled as part of a systematic study of $\delta^{13}$C records; quantitative precipitation reconstructions for the Holocene were then based upon a robust transfer function between the $\delta^{13}$C of modern common millet and precipitation, providing a scientific basis for predicting future climate change and its possible impact.

## 2 Methods
### 2.1 Sampling

All the ancient common millet remains used in this study were found at five archeological sites in the Guanzhong Basin, *i.e.* the BJC, HDP, MN, BN and NS sites (State Cultural Relics Bureau, 1998). Five sections characterized by continuous and well-developed sedimentation were selected for sampling at the BJC, BN, HDP, MN and NS sites. The sampling interval was 10 cm for the BJC and NS sections, and 20 cm for the BN, HDP and MN sections. Forty litre sample bags were filled with sufficient quantities of sedimentary material to screen through a 50-mesh sieve to obtain samples using flotation (Tsuyuzaki, 1994). Different archeological remains were separated in the laboratory after air-drying. Agricultural seeds were identified and picked out under the stereomicroscope, then marked in order according to sampling depth. The numbers of remnant common millet seeds derived from all five sections are listed in Table 1.

### 2.3 Stable $\delta^{13}$C analysis

Stable $\delta^{13}$C composition analyses were carried out on all 67 serial and bulk common millet samples from the five sections, each composed of three to five grains, without lemma. Each sample portion was placed in a beaker and covered with a 1% hydrochloric acid solution to remove any carbonates. The samples were then washed with distilled water to pH >5 and oven dried at 40°C for 24 h. The dried samples were ground in an agate mortar and homogenized, then vacuum-sealed in a quartz tube with copper oxide and silver foil and combusted for at least 4 h at 850°C. The $CO_2$ gas from the combustion tube was extracted and cryogenically purified. The isotopic ratio of the extracted $CO_2$ gas was determined using a MAT-251 gas source mass spectrometer with a dual inlet system at the Institute of Earth Environment, Chinese Academy of Sciences.

All isotope ratios were expressed using the following $\delta$ notation:

$$\delta^{13}C(‰) = [(R_{sample} - R_{std})/R_{std}] \times 1000 \qquad \text{Eq. (1)}$$

The isotopic standard used was Vienna Pee Dee Belemnite (VPDB); analytical precision at the 1r level was reported as 0.2‰.

### 2.3 $^{14}$C dating and age model

AMS $^{14}$C dating was conducted on one charcoal fragment and one charred seed of common millet from the BJC section, five charred seeds each from the HDP and BN sections, one charred seed from the MN section, and three charred seeds from the NS



section. AMS $^{14}$C dating was carried out in the AMS chemistry laboratories at the
Australian Nuclear Science and Technology Organisation (ANSTO) using a STAR
Accelerator. AMS $^{14}$C dates were calibrated using Calib Rev 7.0.4 software and the
INTCAL13 dataset (Reimer *et al.*, 2013). The AMS $^{14}$C dating results (Table 2) show
that the ages of the sampled sections' cultural layers were usually correspondent with
archeological periodization.
On the basis that the depth-based linear interpolation method was not fit for the
dating of cultural layers because of potential disturbance, all common millet remnant
samples were divided into several groups to guarantee at least one dating dataset for
each group, as follows: samples from adjacent depths with close $\delta^{13}$C values were
placed in the same group, allowing a greater difference between each group (One-
factor Analysis of Variance (one-way ANOVA), P<0.05).
**2.4 Quantitative modeling method and data analysis**
The results for $\delta^{13}$C values in the seeds of modern millet grown on the CLP (ref.
Yang and Li, 2015), demonstrated that the $\delta^{13}$C of common millet has a significant
positive correlation with precipitation. In this study, standard major axis regression
analysis (SMA) was applied to establish a regression model between the $\delta^{13}$C of
modern common millet and precipitation during growing seasons. SMA is
appropriate for random samples because, in data statistics analysis, it can take
account of standard deviations in the fit of both independent variables and dependent
variables. Moreover, any implication drawn from the dataset is both intuitionistic
and perspicuous, and therefore superior to ordinary least squares (OLS). Any
gradient acquired by SMA can more scientifically reflect the scaled relation between
two sets of observations; the regression coefficient of data samples can then be
solved by optimizing statistical responses in accordance with logical optimization
criteria. Statistical analyses were conducted using SMATR software (Version 2.0)
(Falster *et al.*, 2006). Other statistical analyses used SPSS 15.0 for Windows and
OriginPro 8.0 software. Unless otherwise stated, differences were considered
statistically significant when P<0.05.
**3 Results**
Carbonized seed remains, sampled from Neolithic cultural layers, have $\delta^{13}$C
values ranging from -11.11‰ to -9.26‰ (Figure 2a), with a mean of -10.23±0.36‰ (n
＝66, SD=±1 σ), eliminating the anomaly value of -8.82‰ analyzed by Boxplot using
SPSS statistical software (Figure 2b). The $\delta^{13}$C composition of modern common
millet from the central and western CLP measured in 2008 ranged from -13.93‰ to -
12.46‰, with a mean of -13.15±0.50‰ (n＝15, SD=±1σ) (Yang and Li, 2015). It can
thus be seen that the $\delta^{13}$C values of common millet remains are more positive than
those of modern seeds by 〜2.92‰.
The $^{13}$C composition of plants results from a combination of carbon isotope
fractionation and source carbon isotope composition. Therefore, $\delta^{13}$C changes in the
atmosphere, as a part of total $CO_2$, are an important factor impacting upon the $\delta^{13}$C
values in plants (Araus and Buxo, 1993). The $\delta^{13}$C values of atmospheric $CO_2$ in the





Holocene, from 11 ka BP to the pre-industrial age, show only a slight change, usually
ranging between -6.1‰ and -6.6‰, with a mean value of -6.4± 0.15‰ (Marino *et al*.,
1992; Leuenberger *et al*., 1992), ～1.8‰ higher than the present-day atmospheric
$CO_2$ $\delta^{13}C$ values of -8.2‰ (Farquhar *et al*., 1989; Keeling and Whorf, 1992; Cuntz,
2011). After correcting for the change in atmospheric $CO_2$ $\delta^{13}C$ (1.8‰), the seed $\delta^{13}C$
values for Holocene millet from the Guanzhong Basin are equivalent to modern plant
values of -12.01‰, and are therefore ～1.12‰ less depleted in $\delta^{13}C$ than modern
seeds (for the t test, t=21.39).

The regression function between $\delta^{13}C$ and precipitation for the common millet
growing season was established using SMA as follows (Figure 3):

$\delta^{13}C$(‰)= 0.0077$P_{gp}$-14.76,$r^2$= 0.56,P＜0.001          Eq.(2)

The function's gradient indicated that the precipitation coefficient was
0.77‰/100 mm, implying that, within physiological adaptation parameters, there
would be a ~0.77‰ increase in $\delta^{13}C$ with a 100 mm increase in precipitation. The
$\delta^{13}C$ values yielded by ancient common millets are slightly higher than those of
modern common millet seeds, suggesting that these ancient plants grew in a much
more humid environment than today's.

Common millet remains from archeological sites were divided into a total of 11
groups (Table 3). Mean $\delta^{13}C$ values for common millet remains were calculated for
each group. Results showed that the minimum value was -10.55±0.16‰, and the
maximum value -9.56±0.09‰, for common millet growing between 7.7 ka BP and 3.4
ka BP. After correcting for the change in the atmospheric $CO_2$ $\delta^{13}C$ (1.8‰), the range
of mean $\delta^{13}C$ values for ancient millet *vis-à-vis* modern plants was between -12.35‰
and -11.36‰. By applying the regression model based on the $\delta^{13}C$ and precipitation
values for modern common millet during its growing season, we were able to extract
paleoprecipitation values for the growing seasons of ancient crops for certain time
periods.

These paleoprecipitation values were reconstructed by applying the $\delta^{13}C_{re}$ values
for the ancient millet to a regression equation (Eq. 2) which expresses the relation
between the $\delta^{13}C$ of common millet and precipitation. The results showed that the
precipitation for the growing seasons ($P_{gs}$) of ancient millet during the period 7.7-3.4
ka BP varied from 240 mm to 477 mm, with a mean of 354 mm (Table 3).

**4 Discussion**
**4.1 The rationale behind using common millet $\delta^{13}C$ for precipitation**
**reconstruction**

Carbon isotope composition of fossilized plant remains is a useful proxy for the
reconstruction of local paleoclimatic changes, especially when using $\delta^{13}C$ values from
plants which experience a single mode of photosynthesis. Common millet grains have
been widely and continuously preserved throughout the Holocene in northern China.
Fossilized millet seeds were generally formed at low temperatures (~250℃) by
baking (Yang *et al*., 2011a), and deposited in strata over long time periods with
limited interaction with the buried environment. The observed $\delta^{13}C$ values of charred
common millet formed at ~250℃ were 0.2‰ lower than those of the source samples,





and much less than the natural variation typically found in wood (Yang *et al*., 2011b).
The $\delta^{13}C$ signatures conserved in carbonized common millet are thus more reflective
of the true environment, without artificial correction.
The Carbon isotope composition of plants ($\delta^{13}C_p$) is affected by both
physiological characteristics and environmental factors. The $\delta^{13}C$ of $C_3$ plants
responds to environmental factors, such as atmospheric $CO_2$ pressure, $O_2$ partial
pressure, temperature, light and precipitation, by dominating the ratio of the
intercellular and ambient partial pressure of $CO_2$ ($c_i/c_a$) with the opening and closing
of leaf stomata (Körner and Diemer, 1987; Körner and Larcher, 1988; Körner *et al*.,
1989; Farquhar *et al*., 1989; Dawson *et al*., 2002). However, the $\delta^{13}C$ of $C_4$ plants
depends not only on $c_i/c_a$ but also on how much $CO_2$ and $HCO_3^-$ in bundle sheath
cells leaks into the mesophyll cells (called leakiness $\varphi$), which is determined by its
physiological characteristics (Hubick *et al*., 1990). When $\varphi$ is larger/smaller than 0.37,
there is a positive/negative correlation between $\delta^{13}C_p$ and $c_i/c_a$ (Ubierna *et al*., 2011).
Under water stress, the $\varphi$ of the common millet, belonging to the NADP-ME
subgroup of $C_4$ plants, is likely larger than 0.37 (Schulze *et al*., 1996; Yang and Li,
2015). This may account for the relation between the $\delta^{13}C$ of common millet and
precipitation being significantly positive (Yang and Li, 2015).
Limited precipitation and soil humidity are the most important environmental
factors affecting the growth of plants in arid and semi-arid areas (IPCC, 2007). For $C_4$
species in the arid regions of northwestern China, $\delta^{13}C_p$ tends to decrease with
decreasing soil water availability (Wang *et al*., 2005b). For common millet, although
altitude, precipitation and water availability have a significant correlation with $\delta^{13}C$
according to correlation analysis, precipitation was the principal control of $\delta^{13}C$,
based on functional mechanism analysis (Yang and Li, 2015). The plants'
physiological characteristics and morphological adaptability showed that the stomatal,
and some non-stamatal, factors of common millet are sensitive to water status,
causing the $\delta^{13}C$ of the organic material to change with precipitation. This rationale
establishes an important theoretical foundation whereby the $\delta^{13}C$ of common millet
can serve as an effective indicator of paleoprecipitation.

**4.2 Comparison between the mid-Holocene and modern precipitation**
Ancient equivalent-seed $\delta^{13}C$ values, ranging from -12.35‰ to -11.36‰, are $\sim$
1–2‰ higher than those for modern millet seeds in the area. However, these analyses
reveal a small but significant shift to lower $\delta^{13}C$ values in modern seeds. Based on the
positive relation between the $\delta^{13}C$ of modern common millet and precipitation in the
CLP, the observed increases in the $\delta^{13}C$ values of ancient millet seeds could have been
caused by increased precipitation. Calculations using the regression equation provide
conservative estimates of the magnitude of this $\delta^{13}C$ shift in precipitation.
The $\delta^{13}C$ values of common millet seeds reflect the $^{13}C$ of photosynthetic
materials during not only their formative and mature stages, but also their vegetative
stage. The growing season of modern common millet in the Guanzhong Basin lasts
from June to September. The seed kernel's formative and mature stages occur soon





after pollination of the blossom. With an increase in kernel size, photosynthetic
material as well as pre-accumulated organic material is transferred to kernels from
stems, leaves and spikes (Chai, 1999). Therefore, millet $\delta^{13}C$ reflects the
environmental conditions extant during the growing season from mid-June to the end
of September, or 110 days in total.
Precipitation data from the Guanzhong Basin for the period 1951-2011 were
analyzed. The results showed that the precipitation for mid-June to September was
between 110-526 mm, with a mean of 305 mm (Figure 4). The 95% confidence
interval for this mean is between 279-332 mm, ruling out the extreme values of
abnormal years. Paleoprecipitation reconstructed from the regression function shows
that growing season precipitation for millet from 7.7-3.4 ka BP was between 240 and
477 mm, with a mean of 354 mm. Summer paleoprecipitation values show that the
climate was much more humid than it is today, with mean precipitation ~50 mm, or
17%, higher. A peak mean summer precipitation of 442 mm was reached at ~5.7 ka
BP; even the lowest value of 313 mm ~6.5 ka BP was higher than today's mean value.
Summer precipitation during the Mid Holocene (7.7-3.4 ka BP) in the Guanzhong
Basin exhibited a systemic increase.
The reconstructed summer precipitation also fluctuates significantly.
Accordingly, the 7.7-3.4 ka BP period can be divided into four distinct stages (Figure
4). During the 7.7-6.4 ka period, summer precipitation was 332 mm, which is 9%, or
27 mm, higher than today. For 6.4-5.5 ka BP, summer precipitation was 414 mm, *i.e.*
109 mm, or 36%, higher than today. During 5.5-4.4 ka BP, summer precipitation was
338 mm, higher than today's value by 33 mm, or 11%. During the period 4.4-3.4 ka
BP, summer precipitation was 361 mm, *i.e.* 18%, or 56 mm, higher than today.
On the basis of the above analysis, the period 6.4-5.5 ka BP, having the most
abundant precipitation and being the most markedly humid period, probably marks
the Holocene Climate Optimum in the Guanzhong Basin; this was also when the
Yangshao Culture flourished, with archeological finds indicating that there were as
many villages in the area as there are today. It is worth noting that the relatively high
precipitation during 4.4-3.4 ka BP was mainly caused by an anomalous high value of
397±11 mm at ~4.1 ka BP, when precipitation was 92 mm, or 30% higher, than at
present. This may indicate a rapidly-developing climatic event, correspondent with
other global records.

**4.3 Validating the reliability of quantitative precipitation reconstructions**

The instrumental data for the last 60 years (1961-2011) indicate that precipitation
in the Guanzhong Basin occurs mainly in the summer (Figure 5). The current inland
flow of warm/humid air dominated by the EASM during the summer (June through
September) delivers ~58% of the total annual precipitation. The area is a typical
monsoon precipitation area, and summer precipitation here is therefore sensitive to
variations in the EASM.
Previous studies of various climatic proxies including stalagmite $\delta^{18}O$, lacustrine
sediments and loess-paleosols all indicate that the CLP had plenty of rain in the
Holocene and was much more humid during the Mid Holocene (Shen *et al.*, 2005;



Wang *et al*., 2005a; Wang *et al*., 2008; Wang *et al*., 2014; Chen *et al*., 2015). The
frequency of paleosol development increased during ~8.6-3.2 ka in the CLP (Wang *et*
*al*., 2014). The eolian-sand activities in the sandlands located to the north of the CLP
decreased from ~8.6-3.2 ka BP (Wang *et al*., 2014; Yang *et al*., 2012), whilst the
vegetation coverage of the desert/loess transitional zone increased in this interval
(Yang *et al*., 2015). These various proxy records infer that the EASM was stronger
during the Mid Holocene, but the amplitude of any variations in the EASM remains
difficult to assess. Fortunately, summer precipitation in northern China provides an
effective approach to determining EASM intensity (Liu *et al*., 2015).
Our quantitative reconstructions of summer precipitation based on millet $\delta^{13}C$
indicate that EASM intensity peaked during 6.4-5.5 ka BP. The strongest summer
monsoon brought the wettest climate, with 36% higher precipitation than today's.
More evidence supporting our contention comes from the tree pollen records from
lake sediments around the CLP, which respond more directly to changes in the EASM
than the other records because trees on the margins of monsoonal regions are sensitive
to variations in monsoonal precipitation. Pollen records from Qinghai Lake, located to
the west of the Guanzhong Basin and on the modern monsoon margins, indicate a wet
interval during 7.4-4.5 ka BP, culminating in a peak at 6.5 ka BP (Figure 6a) (Shen *et*
*al*., 2005). Although the increase in precipitation cannot be assessed, the general trend
is comparable with our $\delta^{13}C$-based precipitation reconstruction results. The percentage
of broadleaf trees from pollen record in the Gonghai Lake (on the northeastern
margins of the CLP; Figure 6b), indicate that the peak monsoonal period occurred
during ~7.8-5.3 ka BP, with an average annual precipitation of 574 mm (Figure 6c),
~30% higher than the modern value (Chen *et al*., 2015). The increase in precipitation
is highly consistent with our reconstruction results. More evidence from PMIP2 (the
second phase of the Paleoclimate Modeling Intercomparison Project) coupled with
Mid Holocene simulations showed that the summer precipitation associated with the
EASM increased throughout most of China ~6 ka BP. The increase in precipitation in
the Guanzhong Basin was lower than 255.5 mm/yr at that time, as inferred from the
greatest increases in precipitation seen in the region, *i.e.* the southern margins of the
Tibetan Plateau, and southeastern coastal area of China, which experienced
precipitation increases of >1.5 mm/day and 0.7 mm/day (or 547.5 mm/yr and 255.5
mm/yr), respectively (Zhang and Liu, 2009). These multiple lines of evidence
corroborate our reconstructions, not only *vis-à-vis* changes in precipitation during the
Holocene, but also their quantitative accuracy.
**5 Conclusions**
Summer precipitation from 7.7 to 3.4 ka BP  reconstructed using the $\delta^{13}C$ values
of common millet was 240-477 mm, with a mean of 354 mm, ～50 mm, or 17%,
higher than at present. Maximum mean summer precipitation peaked at 414 mm,
~109 mm (or 36%) higher than today; this occurred during the period 6.4-5.5 ka BP,
indicating that the EASM peaked at this time.
Although the $\delta^{13}C$-based precipitation record in this study has a low-resolution,
the work provides a convincing method and proxy for establishing the





paleoprecipitation record. Carbonized common millet remains from the Neolithic Age
onward can provide a reliable dating framework and aid the reconstruction of
continuous paleoprecipitation sequences. This, in turn, can allow regional
comparisons, providing a scientific foundation for promoting further research into the
quantitative reconstruction of regional paleoclimates, and helping to understand the
detailed processes and precise mechanisms of the EASM, as well as the relation
between early human activity and environmental change.

**Authorial contributions**

X. Q. L.: overall coordination of writing, sampling, $^{14}$C dating and paleoprecipitation
reconstruction; Q. Y.: writing, sampling, data processing and paleoprecipitation
reconstruction; X. Y. Z. and K. L. Z.: sampling and data processing; N. S.: sampling and $^{14}$C
dating. All authors reviewed the manuscript.

**Acknowledgements**

This study was supported by the National Basic Research Program of China (Grant No.
2015CB953804, 2015CB953803), the National Natural Science Foundation of China (Grant
Nos. 41301042, 41372175) and the National Science Fund for Talent Training in Basic
Science (Grant No. J1210008). We thank Prof. John Dodson for AMS$^{14}$C dating support, and
Dr. Ying Xi for assistance with collecting the original meteorological data.

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



## Tables

**Table 1 Sampling sites and numbers of remnant common millet samples**

| Sites | Cultural types | Source | No. |
|---|---|---|---|
| BJC | Early Laoguantai Culture | | 12 |
| HDP | Banpo type, Yangshao Culture | Cultural layers | 9 |
| MN | Miaodigou type, Yangshao Culture | Cultural layers | 11 |
| BN | Longshan Culture | Cultural layers | 15 |
| NS | Erlitou Culture, Shang Dynasty | Cultural layers | 20 |

**Table 2 AMS$^{14}$C dating data**

| Sample code | Depth (cm) | Sample type | AMS $^{14}$C age (cal yr BP) | Calibrated age range (cal yr BP, 2σ) | Lab code |
|---|---|---|---|---|---|
| BJC-1 | 180-190 | Common millet | 6,705±40 | 7,580±76 | OZM447 |
| BJC-2 | 50-60 | Charcoal | 6,675±40 | 7,543±68 | OZM446 |
| HDP-1 | 235-250 | Common millet | 5,720±50 | 6,523±115 | OZM473 |
| HDP-2 | 160-180 | Rice seed | 5,015±45 | 5,775±120 | OZM472 |
| HDP-3 | 80-100 | Rice seed | 5,120±35 | 5,790±40 | OZM471 |
| HDP-4 | 40-60 | Foxtail millet | 5,185±40 | 5,948±59 | OZM470 |
| MN-1 | 140-160 | Foxtail millet | 4,550±35 | 5,121±70 | OZM452 |
| BN-1 | 280-300 | Foxtail millet | 5,450±70 | 6,286±113 | OZM481 |
| BN-2 | 220-240 | Foxtail millet | 3,820±45 | 4,191±109 | OZM480 |
| BN-3 | 180-200 | Rice seed | 3,770±35 | 4,158±85 | OZM479 |
| BN-4 | 120-140 | Foxtail millet | 4,540±50 | 5,181±141 | OZM478 |
| BN-5 | 40-60 | Common millet | 4,110±40 | 4,625±104 | OZM477 |
| NS-1 | 230-240 | Wheat seed | 3,300±30 | 3,521±70 | OZM460 |
| NS-2 | 200-210 | Wheat seed | 3,280±35 | 3,514±73 | OZM459 |
| NS-3 | 140-150 | Wheat seed | 3,300±30 | 3,521±70 | OZM458 |

**Table 3 Information for grouped common millet remains, by section**

| Section | Cultural age | Depth (cm) | N | $^{14}$C Age (cal yr BP) | Mean $\delta^{13}$C(‰) | $\delta^{13}$C$_{re}$ (‰) | P$_S$ (mm) |
|---|---|---|---|---|---|---|---|
| BJC | Laoguantai Culture | 50-190 | 12 | 7,475-7,656 | -10.36±0.23 | -12.16 | 337±30 |
| HDP | Banpo type, Yangshao Culture | 200-250 | 3 | 6,407-6,638 | -10.55±0.29 | -12.35 | 313±37 |
| | Banpo type, Yangshao Culture | 120-200 | 2 | 5,654-5,895 | -9.56±0.12 | -11.36 | 442±16 |
| | Banpo type, Yangshao Culture | 0-120 | 3 | 5,749-6,008 | -9.87±0.09 | -11.67 | 401±12 |
| MN | Miaodigou type, Yangshao Culture | 0-260 | 11 | 5,052-5,190 | -10.37±0.36 | -12.17 | 336±47 |
| BN | Yangshao Culture | 260-300 | 2 | 6,171-6,399 | -9.84 | -11.64 | 405 |
| | Longshan Culture | 200-260 | 3 | 4,137-4,359 | -10.23±0.04 | -12.03 | 354±5 |
| | Longshan Culture | 140-200 | 3 | 4,072-4,243 | -9.9±0.09 | -11.70 | 397±11 |
| | Longshan Culture | 60-140 | 4 | 5,039-5,322 | -10.28±0.21 | -12.08 | 348±27 |
| | Longshan Culture | 40-60 | 2 | 4,520-4,729 | -10.42±0.07 | -12.22 | 330±9 |
| NS | Erlitou Culture, Shang Dynasty | 50-300 | 20 | 3,444-3,592 | -10.23±0.36 | -12.02 | 356±54 |




## Figures

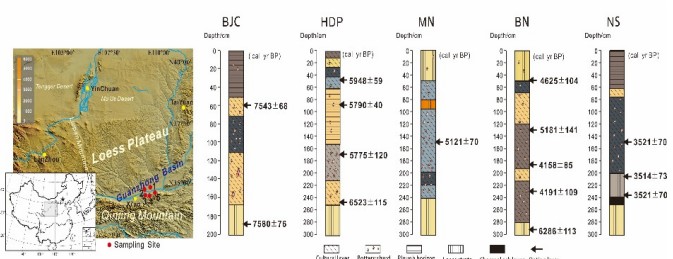

**Figure 1.** Location of sampling sites and description of all sampling sections. Red solid circles indicate sampling sites. 1, Baijiacun site (BJC) (34°33′7.53″N, 109°24′38.6″E); 2, Huiduipo site (HDP) (34°34′4.1″N, 109°01′41.8″E); 3, Manan site (MN) (34°28′23.7″N, 109°05′17.5″E); 4, Beiniu site (BN) (34°28′14.5″N, 109°19′2.6″E); 5, Nansha site (NS)(34°29′30.1″N, 109°42′47.9″E). The map was created using Global Mapper 14.0 software (http://www.skycn.com/soft/appid/11312.html) and NASA STRM data with a resolution of 90 m; profiles were then drawn and combined with the map using CorelDRAW 12 software (http://www.xp85.com/html/CorelDRAW12.html).

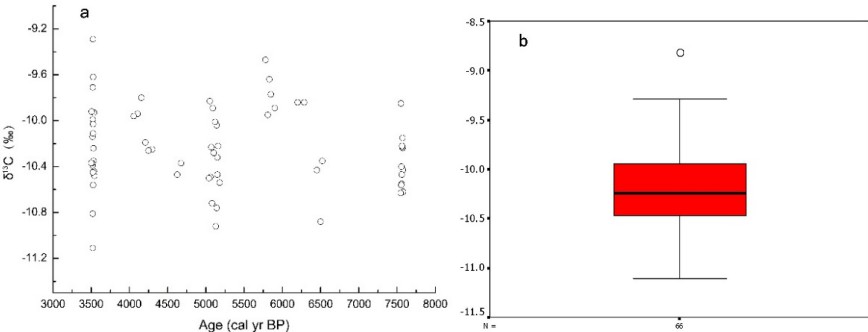

**Figure 2.** $\delta^{13}$C of common millet from archeological sites, Guanzhong Basin





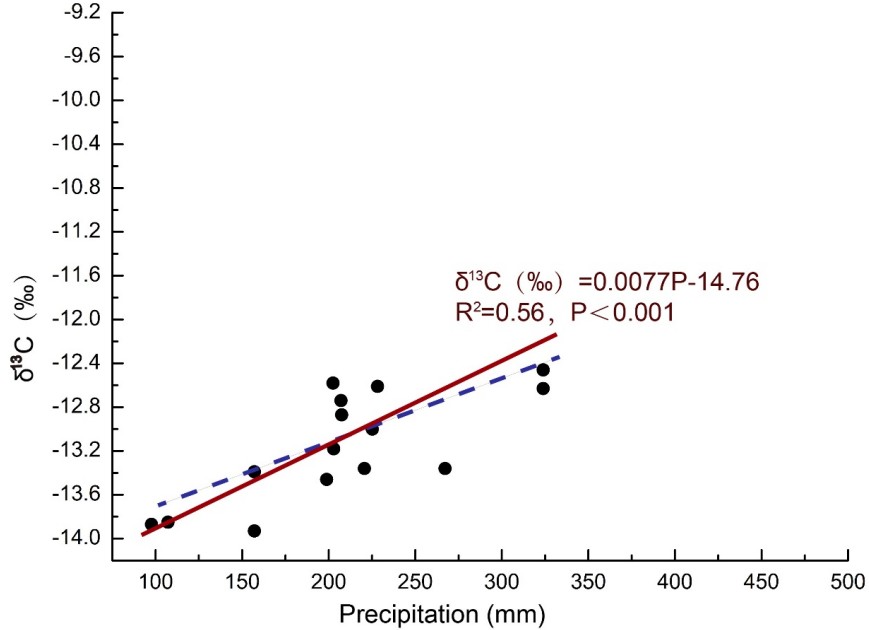

**Figure 3.** The regression model of the $\delta^{13}C$ of modern common millet and summer precipitation.

Dark red line denotes the line of best fit established using SMA; the blue dotted line denotes the

line of best fit established using OLS.

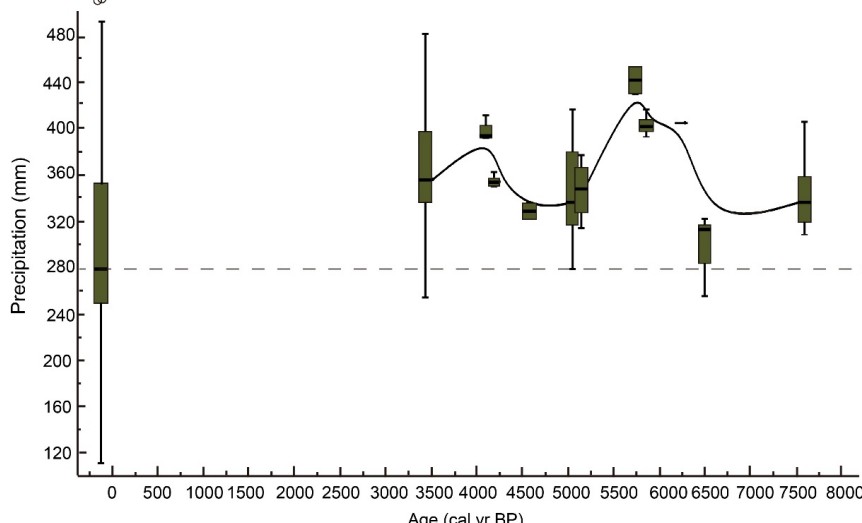

**Figure 4.** Precipitation for mid-June to September for a modern period (original data for 1951–

2011, from the China Meteorological Administration) and 7.7-3.4 ka BP, Guanzhong Basin





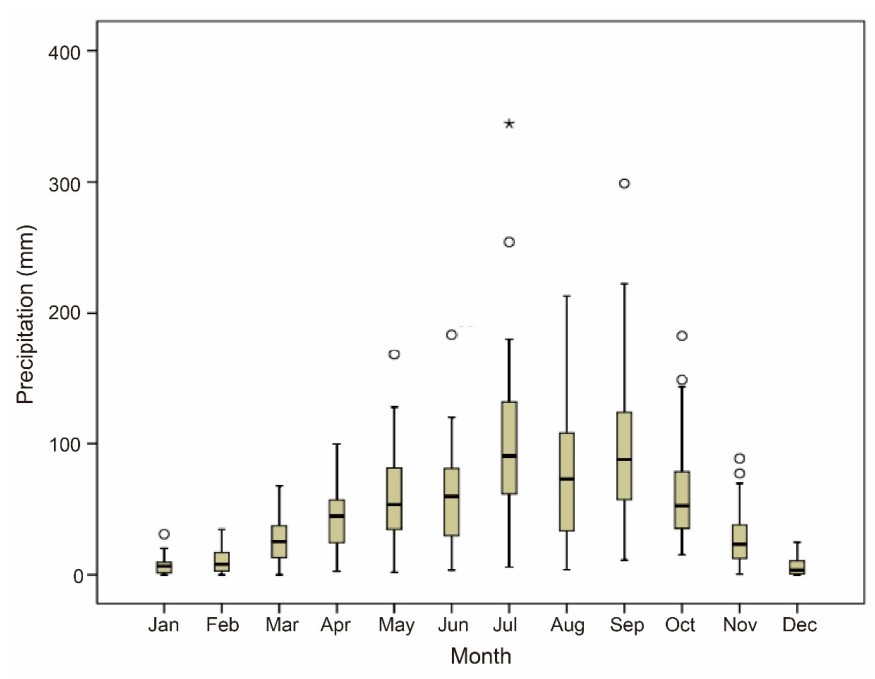

609

**Figure 5.** Instrumental precipitation data for 1951-2011 from Xi'an Station, Sha'anxi, China
(original data, Data Sharing Platform, China Meteorological Administration). The empty circle (○)

indicates an abnormal value; the asterisk (*) indicates an extremely abnormal value.

613





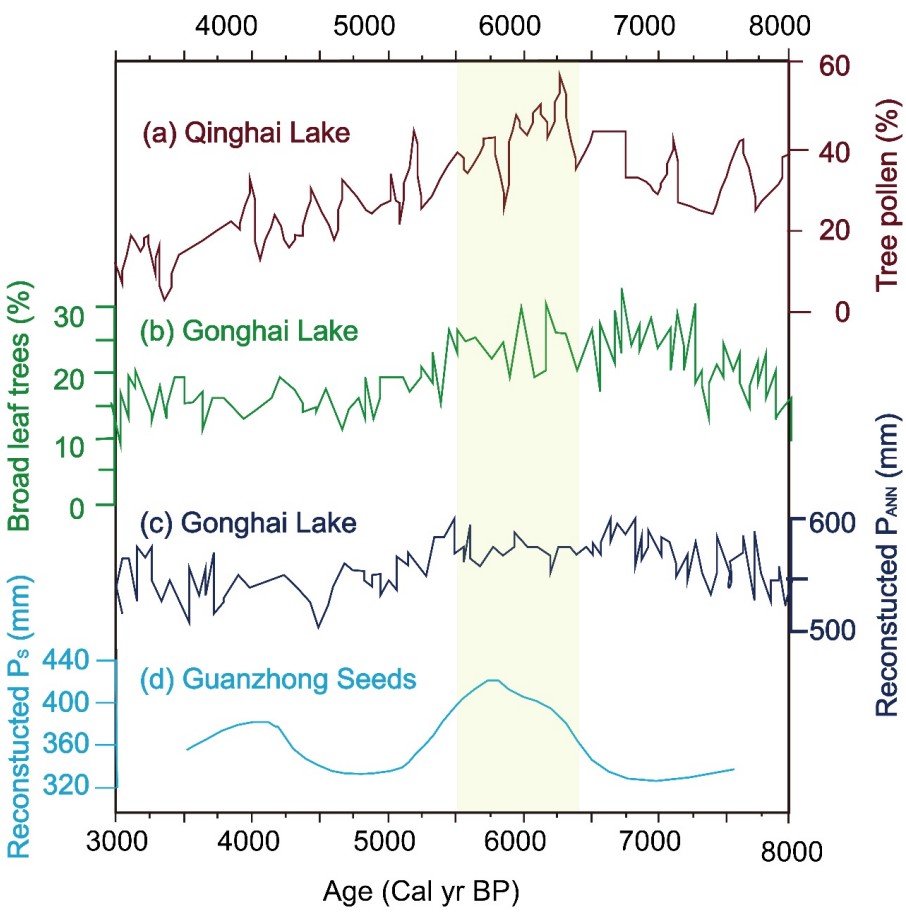

**Figure 6.** Comparison of reconstructed summer precipitation for 7.7-3.4 ka BP, Guanzhong Basin,

with the pollen records of lakes sediments from around the CLP. (**a**) Tree pollen percentages from

Qinghai Lake (Shen *et al*., 2005). (**b**) Broadleaf tree pollen percentages from Gonghai Lake (Chen

*et al*., 2015). (**c**) Reconstructed annual precipitation from the pollen records of Gonghai Lake

(Chen *et al*., 2015). (**d**) Reconstructed summer precipitation from the $\delta^{13}C$ values of common

millet, Guanzhong Basin.