# Peer review of "Quantitative reconstruction of summer precipitation using a Mid"

_Climate of the Past, 2016_

## Referee Comment (RC1) · Anonymous Referee #1 · 29 Sep 2016

general comments

The paper intends to demonstrate the suitability, accuracy and usefulness of d13C of millet seed as proxy of paleoprecipitation. Application is performed for late Holocene in nothern China. This study is innovative and definitively deserves to be published in Climate of the Past. I do not have any irremediable concerns : raw data should be provided and I have some propositions 1- to tone down a little bit the writing to make it closer to the reality, 2- to be more precise in the texte when talking about general concepts, 3- to be more accurate when reporting data by e.g. including uncertainty ranges and by propagating them and 4- to re-organize a little bit the manuscript. See details for these specific comments.

details * line 16: please replace "are highly suited" by "are suited", this is enough * line 40: "modern records", do you mean "instrumental records"? please correct. * lines 52-54: this better suits to late Holocene, even the newly acknowledged Anthropocene. Please be specific * lines 55-58: this is clearly overstatement. Megathermal was under quite different external forcings (insolation, CO2, ..) and can not be considered analog of future climate. This even for impacts as the warming recorded at mid Holocene was not global and the present global warming. This sentence does not furthermore have any added-value. Please remove * line 73: please correct Hetté into Hatté * line 74: please decline EASM * lines 101-102 : This has led [. . .] results. Aggressive and useless. please remove. * line 103: ". . . a continuous distribution.." I don't know here if you're talking "in general" or if you already focus on millet. Pollen records are continuous, that's not the case for millet records. They might be numerous in a sedimentary record, they remain discrete and their absence can be interpreted as both i- too dry to allow millet to growth and to produce seed" or ii- bad luck * lines 109-110: " . . . agricultural rain-fed crop. . .": how can you deal with irrigation? I guess this biaise your signal towards more humid condition. How do you statiscally deal with that issue? * lines 132 and everywhere else: acronyms are OK on figure but please avoid them in the manuscript or restric them to DNA and USA. Nobody will remain what HDP is putting for. Keep the extended name in the manuscript. You don't have words limit! * line 145: please precise "continuous" sampling if you did slice sampling (I understand you did). * line 155: the total in table 1 is 66 not 67 seeds * line 159: what do you mean with "distilled water". I don't know any lab that still distilles water. is it ultra-pure water? reverse osmosis purified water? desionised water? * §2.2.: please complete the table 1 with the following information: how many measurements per site, did you run standard (even home reference) to evaluate the fractionation that can occur all along the different steps? please provide us with the values and varibility on reference (is it the 0.2‰ you mention at the end of the §?). * line 172: only to let me know, why did not you split the millet derived gas into 2 aliquots: one for d13C and one for 14C measurements? you would have had both data on a very homogeneous samples. *

§2.3: please provide us with more information on chemical treatment and reduction prior the 14C physical measurement as you did for 13C. * lines 181-186: - please separate these lines from the preceding, they should be in a §entitled "processing data" or something like that. - please provide us with raw data -> add a figure with all d13C and 14C versus depth and the group you built. - please show us in a figure where are the raw data and what the group you created we really need to understand what you did and what is the rationale behind this ANOVA that allowed you to do so. * lines 193-204: these lines seem to be the result of hard time for authors. It seems they had to fight a lot to impose this SMA. Your choice was acknowledged by the publication of the Yang and Li, 2015 's paper. No need to demonstrate, here again, the appropriatness of the methodological approach. Please remove. * line 207 ". . . Neolithic .." do you mean "all seeds" or do you restrict to some of them. Please specify. That's the first time , you're talking about neolithic * lines 208 and everywhere else: ". . . from -11.11‰ to -9.26‰ . . .". If analytical error is 0.2‰ one digit is enough. The second does not have any signification. * line 209: you eliminated the -8.8‰ value based on statistics. Did you cross with the lab book to check if there is a physical (lab) reason for that? * line 218: "(Araus and Buxo, 1993)", please also refer to original work of Farquhar or O'Leary. They are the real pioneers. * line 222: the 2015 values in Mauna Loa is -8.5‰ (http://www.esrl.noaa.gov/gmd/obop/mlo/summary.html) please adapt your calculation. Mauna loa is an island, bare and far from any human activities. It was chosen to reflect the global CO2 free from any local impact (human, vegetation). You are not is this configuration and should include the local effect within your estimation. Your database was designed and completed in 2015 in agricultural regions fully impacted by vegetation and human CO2 emission. You were not in a free zone as Mauna Lo and likely your modern millet did growth in a much more negative atmospheric CO2 that you think. Please discuss this point and (if possible) add d13C measured on modern atmospheric CO2 sampled in locations you collected modern seeds to evaluate the modern shift between Mauna Loa and the CO2 modern millet used for photosynthesis. * line 228 ". . . growing season . . . " should be defined.. but will be defined if you follow

my proposition to move up a part you presently have in discussion (see lines 302-310)
* line 229 : - what is the subscript "gp" for? - can you provide us with error margin on a
(0.0077) and b (-14.56)? * line 238 "-10.55±0.16‰; the very low value of uncertainty
clear sems to show that you didn't propagate analytical uncertainties to the mean d13C
of each group * line 246: what is the subscript "re" for? * §4.1 should be better just
after the introduction, it is not part of the discussion but part of rationale behind the
approach. This can be part of a "rationale" §with lines 302-310. * line 259: what is the
biblio reference that attests that archeo combustion was performed at temperature of
about 250°C? please add. * line 263-265: only accusations that do not bring any added
value to the paper. Please remove and only keeep "The d13C signatures conserved in
carbonized common millet are thus reflective of the true environment". * line 266: car-
bon without capital letter * line 273: do you mean concentration of $CO_2$ and $HCO_3^-$?
please correct * line 282: instead of IPCC reference, consider the vegetal physiology
original bibliography * line 289: corect stamatal into stomatal * line 296-301: already
stated in results, no need to repeat. remove * line 302-310: move up in a "rationale"
§between intro and methodology * line 311-314: should better belong to methodology,
in site description * line 323 and following: as the absolute value is highly dependent
of the d13C value of the atmospheric $CO_2$ you had for the reference equation, please
consider to discuss relative values: this period of Holocene was wetter or drier than
the other part of Holocene * line 338: please provide references for ".. other global
records". * lines 357-358: no interest, remove * line 361: please be more specific, you
don't have here the wettest climate but the wettest millet growth season. * line 377:
please add a reference for PMIP2 and this specific result

* tables: legends are much too short. please extend them. Table and associated legend
should have a stand-alone value. * table 1: what do you mean with "sources"? please
replace "N°" by "number of grains", replace accronyms by extended names (or define
in legend) * table 2: - replace accronyms by extended names (or define in legend), - I
guess what you call "AMS 14C age (cal yr BP)" is conventional 14C age, thus replace
the column title by "conv. 14C age (yr BP) – 1sigma", - calibrated age range can not be

presented as mean value of range extrema $\pm$ the half-distance between range extrema. This only because the mid point of the interval is not associated to the maximum of probability. Please follow the 14C convention and provide us with the range(s) and the associated probabilty density (yes, for this period of time you might have several intervals that share the 100% of the 2-sigma probability density. You might consider to add the age with the maximum of probability (last column of the IntCal output table) if it better suits to you. * table 3: - replace accronyms by extended names (or define in legend), - in legend, please specify what N and d13Cre are for. - instead of mean d13C provide us with d13C range or add another column - please respect the significance of digits and provide d13C with only one digit

* Figure 1: - make sure sites are visible and add their names (or acronyms) on the figure. - if possible add also the sites you mention in Figure 6 (if not possible, add a map with sites in Figure 6 itself) - please add a sign (star, point, arrow, ..) to show depths the seeds were extracted from (entlarge the figure if required) - this question is maybe more for publisher: is it require to provide references for CorelDraw or others Word or Excel? * Figure 6: the sites mentioned here should be geographically visible in a map, here or on the Figure 1 map. It would be great to locate them within a meteorological context, can you consider to add a limit of monsoon influenced zone?

---

## Referee Comment (RC2) · Anonymous Referee #2 · 1 Oct 2016

The authors presented carbon isotope data from fossilized common millet seeds collected from archeological sites in northern China. The carbon isotope record was then used to reconstruct summer precipitation during the mid-Holocene, on the basis of the relationship between millet C isotopes and summer precipitation established in a modern process study by the same group (Yang and Li 2015). The authors then used the reconstructed summer precipitation to discuss East Asian summer monsoon dynamics.

I find the idea is intriguing and potentially promotes the use of abundantly available millet seeds as a paleoclimate archive in this part of the world. The modern process investigation as presented in Yang and Li (2015) is an excellent study that shows a robust relationship between d13C and summer precipitation, despite that I don't fully understand the mechanism (it is counterintuitive that lower summer precipitation correlates with lower d13C values).

However, I have issues with both presentation and interpretation of the results.

1. I find that the "conventional" climate reconstruction and interpretation as presented in Figure 4 are unsupported, mostly due to the high variability of the record (including the instrumental climate data!), low resolution, and short, snapshot nature of the record (only cover 8-3.5 ka). The high precipitation variability at present in the study region suggests that 3-5 seeds used in each analysis just captured at most 3 to 5 years of precipitation – too small a sample size to capture mean precipitation. As a result, I don't think the correlation as presented in Figure 6 and monsoon discussion is supported.

However, I wonder if that the data set (especially an expanded data set form the region) can be used to document and understand the summer precipitation variability during the Holocene or part of the Holocene. The science question could be: is there increasing summer precipitation variability from the early Holocene to late Holocene, when the summer monsoon and precipitation decline during that time period? Is it possible that not only summer precipitation decreases during the Holocene but also becomes more and more variable and less and less predictable? The data as presented in Fig. 2a seem to suggest that, though the number of analysis is still low. I wonder if a future expanded study can analyze a larger number of samples per sample (say 30-40 seeds, preferably single seed analysis) to capture the decadal/centennial (depending time resolution) variability in summer precipitation, even just in a few time intervals (early Holocene, mid-Holocene, and late Holocene). Each individual seed is a product of a single season/year – as clearly argued and implemented during 2008 in Yang and Li (2015). This is similar to a study on oxygen isotope analysis of individual foraminifera from deep marine sediments to document ENSO variability (and annual seasonal cycle) at a few time intervals during the Holocene.

I suggest that the authors should focus on the variability rather than mean climate

(precipitation).

2. The writing in general is clear – I commend the authors' effort to make it an easy read. However, I find there are many superlative words to describe the results, and some of these are overstatement. I will provide examples below in my specific comments.

Specific comments: Title: -focus on precipitation variability? -change "China" to "northern China", for international readership?

Abstract -add latitude (34.5 N) and perhaps rounded longitude as well, for international readership?

-there are many superlative descriptors here in the abstract, such as "accurate" (line 17), "robust" (line 19), "reliably" (line 26), "precise" (line 28). It seems to me none of these is needed and justified. Most of similar words should be deleted throughout the text.

-the abstract needs to refocus if the authors accept my suggestion above.

Introduction -It is unnecessarily too long. In particular, the general discussion on Holocene climate in the first 3 paragraphs on page 1-2 is not really needed. Delete or shorten.

-superlative word examples: "accurate" (l 41), "more completely and accurately" (l 60), "robust" (l 135).

Line 91: change "between 5.2-4.3 ka BP", to either "between 5.2 AND 4,3 ka BP", or "at 5.2-4.3 ka BP" (there are other cases of matching "between...and..." in the text)

Methods This section reads well.

Line 156: I wonder if a single seed is large enough for C isotope analysis, but multiple seed analysis still can be used for the variability study as suggested above (but it will be "conservative" reconstruction of precipitation variability, due to averaging of multiple

years growth in one sample).

L 169: "1r" = 1 sigma? (67% probability?)

L 179: change to "the sampled culture layers" ("section's" is awkward usage)

L 181-186: unclear how it was done.

L 189: delete "ref."

L 190: delete "," before "demonstrated"

Results L 209: change "eliminating" to "without considering"

Line 207-214: I'm confused here. You describe carbonized Neolithic seed remains and modern common millet, but you compare "modern seeds" in the last sentence. Also, millet is more negative than seeds, rather than "positive" as described. Check.

L 233-235: "slightly higher" and "a much more humid" is contradictory. Overstatement/over-interpretation?

Discussion As I commented above, the mean precipitation reconstruction doesn't allow for much comparison and discussion on summer monsoon, while precipitation variability is potentially a novel aspect of paleoclimate research. Although your current data are not robust enough, it seems to me that it holds great promise for the future project: even just 3 or 4 horizons, with large analysis per horizon.

L 341: change "1961-2011" to "1951-2011"? which makes 60 years and also is consistent from description earlier.

Conclusions L 392-393: "low resolution" and "convincing" are contradictory.

Tables Table 1 -move latitudes and longitudes from Figure 1 to new columns here

-what "source" means here? "12" here means "12 culture layers"? if so, spell out.

-change "No." to lower case and italic "n" (to indicate number of analyses or samples)

Table 2 -Change heading to " AMS 14C dates" ("dating data" is unusual)

-why does a 250-cm-long section (such as BN) have the same or reversed ages? Very rapid accumulation of these layers? I hope it is discussed elsewhere in archeological literature.

-change the heading of column 4 to "AMS 14C date (yr BP)" – it is wrong to say 14C date as "cal yr BP"

-maybe a footnote to indicate the dating lab for OZM

Table 3 -maybe indicate ages as "calibrated ages" to avoid confusion

-add footnote to indicate "d13Cre" and "Ps"

Figure 1. -move latitude and longitude to Table 1

-CorelDRAW12 is not needed to mention, as it is just a map.

Figure 2. -need more explanation about panel a (raw data points) and b (box plot) in figure caption

-again Fig. 2a kind of shows increase in precipitation variability from 8 ka to 3 ka. Have you tried a regression of all the data to see if there is a significant decline (in precipitation) during that period as well? (perhaps the number of data points are still low)

Figure 3 -indicate reference in figure caption "Modified or data from Yang and Li (2015)"

Figure 4 -Again, I don't think it is a good way to present the data as groups to get mean climate/precipitation – considering the large variability almost nothing can be concluded here (a.k.a. the pattern is not robust/convincing, because of uncertainty). See my comments above.

---

## Author Comment (AC1) · 24 Oct 2016

Dear reviewer: We would like to express our feelings of appreciations to you for your kindly help and professional comments to our manuscript entitled "Summer precipitation reconstructed quantitatively using a Mid Holocene $\delta$13C common millet record from Guanzhong Basin, China". We have tried our best to modify the weakness and flaws pointing out by you. Now, we believe that we made a better work which would probably satisfy the reviewer and suitable to be published. The answer to the comments is listing in the following paragraph.

Thanks again for your help. Best wishes!

Sincerely,

Yang Qing and Xiaoqiang Li

Reviewer 1:

General comments The paper intends to demonstrate the suitability, accuracy and usefulness of d13C of millet seed as proxy of paleoprecipitation. Application is performed for late Holocene in northern China. This study is innovative and definitively deserves to be published in Climate of the Past. I do not have any irremediable concerns: raw data should be provided and I have some propositions 1- to tone down a little bit the writing to make it closer to the reality, 2- to be more precise in the text when talking about general concepts, 3- to be more accurate when reporting data by e.g. including uncertainty ranges and by propagating them and 4- to re-organize a little bit the manuscript. See details for these specific comments. Details * line 16: please replace "are highly suited" by "are suited", this is enough Thanks for the reviewer's suggestion. We have removed "highly" from the sentence.

*Line 40: "modern records", do you mean "instrumental records"? Please correct. Thanks for the reviewer's suggestion. We have corrected "modern records" into "instrumental records" following the suggestion.

* Lines 52-54: this better suits to late Holocene, even the newly acknowledged Anthropocene. Please be specific Thanks for the reviewer's suggestion. According to the two reviewers' suggestions, we have shorten the first three paragraphs and this sentence have been removed.

* Lines 55-58: this is clearly overstatement. Megathermal was under quite different external forcings (insolation, CO2, ..) and can not be considered analog of future climate. This even for impacts as the warming recorded at mid Holocene was not global and the present global warming. This sentence does not furthermore have any added-value. Please remove Thanks for the reviewer's suggestion. According to

the two reviewers' suggestions, we have shorten the first three paragraphs and this sentence have been removed.

*Line 73: please correct Hetté into Hatté Thanks for the reviewer's kind remind. We have corrected Hetté into Hatté..

* Line 74: please decline EASM Thanks for the reviewer's suggestion. Considering the integrity and coherence of the manuscript, we have revised the paragraph, adding the research significance of precipitation in the CLP rather than declined EASM, hoping EASM appears in the appropriate place.

*Lines 101-102: This has led [: : :] results. Aggressive and useless. Please remove. Thanks for the reviewer's suggestion. We have removed the sentence following the suggestion.

* Line 103: ": : : a continuous distribution.." I don't know here if you're talking "in general" or if you already focus on millet. Pollen records are continuous, that's not the case for millet records. They might be numerous in a sedimentary record, they remain discrete and their absence can be interpreted as both i- too dry to allow millet to growth and to produce seed" or ii- bad luck Thanks for the reviewer's suggestion. According to the suggestion, we have removed "a continuous distribution" from the sentence to avoid confusion.

* Lines 109-110: " : : : agricultural rain-fed crop: : :": how can you deal with irrigation? I guess this bias your signal towards more humid condition. How do you statistically deal with that issue? Thanks for the reviewer's question. First, common millet is a typical agricultural rain-fed crop. Irrigation in favor of plant growth but the yield of seed will decrease. Secondly, the exploring model to distinguish carbon isotope composition of crops derived from natural precipitation or irrigation has been put forward by Ferrio et al. (2005). According to the references Yang and Li (2015) and Ferrio et al( 2005), we inferred the abnormal high value probably indicate more water supply. So we excluded the abnormal high value according to the Boxplot using SPSS statistical

software. References: Ferrio J P, Araus J L, Buxò R, et al. Water management practices and climate in ancient agriculture: inference from the stable isotope composition of archaeobotanical remains, 2005, 14: 510-517. Yang, Q., and Li, X. Q. Investigation of the controlled factors influencing carbon isotope composition of foxtail and common millet on the Chinese Loess Plateau, Sci. China Ser D, 58(12), 2296-2308, 2015.

* Lines 132 and everywhere else: acronyms are OK on figure but please avoid them in the manuscript or restric them to DNA and USA. Nobody will remain what HDP is putting for. Keep the extended name in the manuscript. You don't have words limit! Thanks for the reviewer's suggestion. According to the suggestion, we have extended all the acronyms for the full names in the manuscript.

* Line 145: please precise "continuous" sampling if you did slice sampling (I understand you did). Thanks for the reviewer's suggestion. We really did slice sampling, so following the suggestion, the sentence was changed into "The slice sampling were applied to continuously sampling and the interval was . . .. . .".

* Line 155: the total in table 1 is 66 not 67 seeds Thanks for the reviewer's attention. The total samples for $\delta 13C$ analysis is really 67 seeds here, but there is one abnormal value which was excluded in the subsequent table.

* Line 159: what do you mean with "distilled water". I don't know any lab that still distills water. is it ultra-pure water? reverse osmosis purified water? deionized water? Thanks for the reviewer's question. It was deionized water. To be more specific, we have corrected "distilled water" into "deionized water" in the manuscript.

* §2.2.: please complete the table 1 with the following information: how many measurements per site, did you run standard (even home reference) to evaluate the fractionation that can occur all along the different steps? please provide us with the values and variability on reference (is it the 0.2‰ you mention at the end of the §?). Thanks for the reviewer's suggestion. The column of n means the number of measurements per site. We have revised the table and note the meaning of n. The fractionation that can occur

all along the steps is the 0.2‰ as we mentioned at the end of the paragraph.

* Line 172: only to let me know, why did not you split the millet derived gas into 2 aliquots: one for d13C and one for 14C measurements? you would have had both data on a very homogeneous samples. Thanks for the reviewer's question. Because the millet individuals are very tiny and a single millet is even not enough for the $\delta$13C measurement, three to five grains were composed for $\delta$13C analysis. That's why we cannot split the millet derived gas into 2 aliquots: one for $\delta$13C and one for $\delta$14C measurement.

* §2.3: please provide us with more information on chemical treatment and reduction prior the 14C physical measurement as you did for 13C. Thanks for the reviewer's suggestion. We have added the brief introduction on chemical treatment and reduction prior the 14C physical measurement in the manuscript.

* lines 181-186: - please separate these lines from the preceding, they should be in a "processing data" or something like that. - please provide us with raw data -> add a figure with all d13C and 14C versus depth and the group you built. - please show us in a figure where are the raw data and what the group you created we really need to understand what you did and what is the rationale behind this ANOVA that allowed you to do so. Thanks for the reviewer's suggestion. We have separated the lines as another section entitled "processing data of age model" and added a figure (Figure 3a) with all $\delta$13C and calibrated age range versus depth as well as the groups we built following your suggestion, hoping the readers can understand what we have done and why we did so.

* lines 193-204: these lines seem to be the result of hard time for authors. It seems they had to fight a lot to impose this SMA. Your choice was acknowledged by the publication of the Yang and Li, 2015 's paper. No need to demonstrate, here again, the appropriatenessb of the methodological approach. Please remove. Thanks for the reviewer's suggestion. We have removed the related content according to the suggestion.

* line 207 ": : : Neolithic .." do you mean "all seeds" or do you restrict to some of them. Please specify. That's the first time , you're talking about neolithic Thanks for the reviewer's suggestion. To be more specific, we have modified the sentence into "Common millet remains sampled from cultural layers of Guanzhong Basin in our study…….".

* lines 208 and everywhere else: ": : : from -11.11‰ to -9.26‰ : : :". If analytical error is 0.2‰ one digit is enough. The second does not have any signification. Thanks for the reviewer's suggestion. We have modified the related content and kept all per mil numerical value one digit left.

* line 209: you eliminated the -8.8‰ value based on statistics. Did you cross with the lab book to check if there is a physical (lab) reason for that? Thanks for the reviewer's question. We did cross with the lab book to check this abnormal value, but no runtime exception occurred and the sample was not contaminated. According to the references Yang and Li (2015) and Ferrio et al( 2005), we inferred the plant of sample probably grew in a good ground upon many waters. In this situation, it cannot be included for precipitation reconstruction. References: Ferrio J P, Araus J L, Buxò R, et al. Water management practices and climate in ancient agriculture: inference from the stable isotope composition of archaeobotanical remains, 2005, 14: 510-517. Yang, Q., and Li, X. Q. Investigation of the controlled factors influencing carbon isotope composition of foxtail and common millet on the Chinese Loess Plateau, Sci. China Ser D, 58(12), 2296-2308, 2015.

* line 218: "(Araus and Buxo, 1993)", please also refer to original work of Farquhar or O'Leary. They are the real pioneers. Thanks for the reviewer's suggestion. We have added the references of Farquhar (1989) and O'Leary (1988).

* line 222: the 2015 values in Mauna Loa is -8.5‰ (http://www.esrl.noaa.gov/gmd/obop/mlo/summary.html) please adapt your calculation. Mauna loa is an island, bare and far from any human activities. It was chosen

to reflect the global CO2 free from any local impact (human, vegetation). You are not is this configuration and should include the local effect within your estimation. Your database was designed and completed in 2015 in agricultural regions fully impacted by vegetation and human CO2 emission. You were not in a free zone as Mauna Lo and likely your modern millet did growth in a much more negative atmospheric CO2 that you think. Please discuss this point and (if possible) add d13C measured on modern atmospheric CO2 sampled in locations you collected modern seeds to evaluate the modern shift between Mauna Loa and the CO2 modern millet used for photosynthesis. Thanks for the reviewer's suggestion. Here, authors would like to say: the modern millet was sampled in 2008 rather than in 2015. Although we don't have data of $\delta$13C measured on modern atmospheric CO2 sampled in locations where we collected modern seeds, considering our atmosphere is a perfect blender, we adopted the global mean value of three years after sampling, just as we used the mean value for the past period, from 11 ka BP to the pre-industrial age. So, we consider that the value -8.2‰ which published by Cuntz in 2011 should be more appropriate, even though the samples grew in agricultural regions but not in a free zone. If we adopt the value -8.5‰ to calculate, the reconstructed results would be amplified and bias the environment towards more humid. Based on the above consideration, we didn't adopt the reviewer's suggestion on this issue and hope the reviewer understanding.

* line 228 ": : : growing season : : : " should be defined.. but will be defined if you follow my proposition to move up a part you presently have in discussion (see lines 302-310) Thanks for the reviewer's suggestion. We have moved up the lines 302-310 to just after the introduction and "growing season" has been defined in this section as follow: The growing season of modern common millet in the Guanzhong Basin lasts from June to September.

* line 229 : - what is the subscript "gp" for? Thanks for the reviewer's carefulness. "gp" is short for "growing period", but to keep the internally consistent within the manuscript, we have changed "gp" into "gs".

- can you provide us with error margin on a (0.0077) and b (-14.56)? Thanks for the reviewer's question. But we are sorry to say we cannot provide error margin on a (0.0077) and b (-14.56) since the SMATR software doesn't provide the margin. However, the regression coefficient of data samples are optimized which were solved by optimizing statistical responses in accordance with logical optimization criteria.

* line 238 "-10.55_0.16‰Í , the very low value of uncertainty clear seems to show that you didn't propagate analytical uncertainties to the mean d13C of each group Thanks for the reviewer's enquiry. The values are close to each other in each individual group, so the uncertainty is assuredly the very low value in the group.

* line 246: what is the subscript "re" for? Thanks for the reviewer's question. It means corrected value for precipitation reconstruction. To avoid confusion, we change "$\delta$13Cre" into "corrected $\delta$13C".

* §4.1 should be better just after the introduction, it is not part of the discussion but part of rationale behind the approach. This can be part of a "rationale" lines 302-310. Thanks for the reviewer's suggestion. According to your suggestion, we have moved up the lines 302-310 to just after the introduction and entitled "2 The rationale behind using common millet $\delta$13C for precipitation reconstruction".

* line 259: what is the biblio reference that attests that archeo combustion was performed at temperature of about 250_C? please add. Thanks for the reviewer's question. The reference is Yang et al. 2011a, which were there in the manuscript.

* line 263-265: only accusations that do not bring any added value to the paper. Please remove and only keep "The d13C signatures conserved in carbonized common millet are thus reflective of the true environment". Thanks for the reviewer's suggestion. We have removed the value and changed the sentence following the suggestion.

* line 266: carbon without capital letter Thanks for the reviewer's kind remind. We have changed carbon without capital letter.

* line 273: do you mean concentration of CO2 and HCO3-? please correct Thanks for the reviewer's question. Here "how much CO2 and HCO3-" expresses more accurate than concentration, and we consider it is more appropriate. So we didn't change it, hoping the reviewer understanding.

* line 282: instead of IPCC reference, consider the vegetal physiology original bibliography Thanks for the suggestion. We have instead the reference by "Hadley and Szarek, 1981; Ehleringer and Mooney, 1983; Murphy and Bowman, 2009".

* line 289: corect stamatal into stomatal Thanks for the reviewer's kind remind. We have corrected it.

* line 296-301: already stated in results, no need to repeat. remove Thanks for the reviewer's suggestion. We have removed them.

* line 302-310: move up in a "rationale" intro and methodology Thanks for the reviewer's suggestion. We have moved up them after the introduction and entitled "2 The rationale behind using common millet $\delta$13C for precipitation reconstruction".

* line 311-314: should better belong to methodology, in site description Thanks for the reviewer's suggestion. We have added site description in the section of sampling and moved line 311-314 to this section.

* line 323 and following: as the absolute value is highly dependent of the d13C value of the atmospheric CO2 you had for the reference equation, please consider to discuss relative values: this period of Holocene was wetter or drier than the other part of Holocene Thanks for the reviewer's suggestion. We have removed the absolute value and added discussion about the increasing variability of summer precipitation from early Holocene to late Holocene and provided the markedly humid periods in the manuscript.

* line 338: please provide references for ".. other global records". Thanks for the reviewer's suggestion. We have added references Cullen and DeMenocal, (2000),

Mayewski et al. (2004) and Wu and Liu (2004) for ".. other global records".

* lines 357-358: no interest, remove Thanks for the reviewer's suggestion. We have moved the sentence following the suggestion.

* line 361: please be more specific, you don't have here the wettest climate but the wettest millet growth season. Thanks for the reviewer's suggestion. We have changed the wettest climate into the wettest millet growth seasons.

* line 377: please add a reference for PMIP2 and this specific result Thanks for the reviewer's suggestion. The reference for PMIP2 is Zhang and Liu (2009) and the specific result is demonstrated in the following sentence. To avoid confusion, we have adjust the sentence as follow: . . . . . .throughout most of China ∼6 ka BP and the greatest increases in precipitation seen in the region,. . . . . . (Zhang and Liu, 2009). According to the result, it can be inferred. . . . . .".

* tables: legends are much too short. please extend them. Table and associated legend should have a stand-alone value. Thanks for the reviewer's suggestion. We have extended legends and given them stand-alone value in each table.

* table 1: what do you mean with "sources"? please replace "N_" by "number of grains", replace accronyms by extended names (or define in legend) Thanks for the reviewer's question and suggestion. The "sources" means "sample source" and we have added "sample" before "sources". We replaced "No." by "n" and gave a footnote "n means number of remnant common millet samples derived from the section." We also replaced accronyms by extended names.

* table 2: - replace accronyms by extended names (or define in legend), - I guess what you call "AMS 14C age (cal yr BP)" is conventional 14C age, thus replace the column title by "conv. 14C age (yr BP) – 1sigma", - calibrated age range can not be presented as mean value of range extrema _ the half-distance between range extrema. This only because the mid point of the interval is not associated to the maximum of probability.

Please follow the 14C convention and provide us with the range(s) and the associated probabilty density (yes, for this period of time you might have several intervals that share the 100% of the 2-sigma probability density. You might consider to add the age with the maximum of probability (last column of the IntCal output table) if it better suits to you. Thanks for the reviewer's suggestion. We have defined the accronyms in the title, changed "AMS 14C age (cal yr BP)" to "Radiocarbon age (14C yr BP)" and changed calibrated age range (cal yr BP, $2\sigma$) into the age interval.

* table 3: - replace accronyms by extended names (or define in legend), - in legend, please specify what N and d13Cre are for. - instead of mean d13C provide us with d13C range or add another column - please respect the significance of digits and provide d13C with only one digit Thanks for the reviewer's suggestion. We have replaced accronyms by extended names, replaced "N" and "$\delta$13Cre" by "n" and "corrected $\delta$13C" respectively, which were defined in footnote. We also provided $\delta$13C range in the column of corrected $\delta$13C.

* Figure 1: - make sure sites are visible and add their names (or acronyms) on the figure. - if possible add also the sites you mention in Figure 6 (if not possible, add a map with sites in Figure 6 itself) - please add a sign (star, point, arrow, ..) to show depths the seeds were extracted from (entlarge the figure if required) - this question is maybe more for publisher: is it require to provide references for CorelDraw or others Word or Excel? Thanks for the reviewer's suggestion. We have added all sites names as well as the sites we mentioned in Figure 6 to Figure 1. We also added signs for sampling depths with triangle in the description of all sampling sections of Figure 1.

* Figure 6: the sites mentioned here should be geographically visible in a map, here or on the Figure 1 map. It would be great to locate them within a meteorological context, can you consider to add a limit of monsoon influenced zone? Thanks for the reviewer's suggestion. We have added a China map with a limit of monsoon influenced zone. The modern Asian summer monsoon limit is shown by a dashed line in the map, where Qinghai Lake, Gonghai Lake and Guanzhong Basin are signed with red dot.

Please also note the supplement to this comment:
http://www.clim-past-discuss.net/cp-2016-87/cp-2016-87-AC1-supplement.pdf

---

## Author Comment (AC2) · 24 Oct 2016

Dear reviewer: We would like to express our feelings of appreciations to you for your kindly help and professional comments to our manuscript entitled "Summer precipitation reconstructed quantitatively using a Mid Holocene $\delta$13C common millet record from Guanzhong Basin, China". We have tried our best to modify the weakness and flaws pointing out by you. Now, we believe that we made a better work which would probably satisfy the reviewer and suitable to be published. The answer to the comments is listing in the following paragraph.

Thanks again for your help. Best wishes!

[Figure]

Sincerely,

Yang Qing and Xiaoqiang Li

Reviewer 2:

The authors presented carbon isotope data from fossilized common millet seeds collected from archeological sites in northern China. The carbon isotope record was then used to reconstruct summer precipitation during the mid-Holocene, on the basis of the relationship between millet C isotopes and summer precipitation established in a modern process study by the same group (Yang and Li 2015). The authors then used the reconstructed summer precipitation to discuss East Asian summer monsoon dynamics. I find the idea is intriguing and potentially promotes the use of abundantly available millet seeds as a paleoclimate archive in this part of the world. The modern process investigation as presented in Yang and Li (2015) is an excellent study that shows a robust relationship between d13C and summer precipitation, despite that I don't fully understand the mechanism (it is counterintuitive that lower summer precipitation correlates with lower d13C values). However, I have issues with both presentation and interpretation of the results. 1. I find that the "conventional" climate reconstruction and interpretation as presented in Figure 4 are unsupported, mostly due to the high variability of the record (including the instrumental climate data!), low resolution, and short, snapshot nature of the record (only cover 8-3.5 ka). The high precipitation variability at present in the study region suggests that 3-5 seeds used in each analysis just captured at most 3 to 5 years of precipitation – too small a sample size to capture mean precipitation. As a result, I don't think the correlation as presented in Figure 6 and monsoon discussion is supported. However, I wonder if that the data set (especially an expanded data set form the region) can be used to document and understand the summer precipitation variability during the Holocene or part of the Holocene. The science question could be: is there increasing summer precipitation variability from the early Holocene to late Holocene, when the summer monsoon and precipitation decline during that time period? Is it possible that not only summer precipitation decreases during the Holocene

but also becomes more and more variable and less and less predictable? The data as presented in Fig. 2a seem to suggest that, though the number of analysis is still low. I wonder if a future expanded study can analyze a larger number of samples per sample (say 30-40 seeds, preferably single seed analysis) to capture the decadal/centennial (depending time resolution) variability in summer precipitation, even just in a few time intervals (early Holocene, mid-Holocene, and late Holocene). Each individual seed is a product of a single season/year – as clearly argued and implemented during 2008 in Yang and Li (2015). This is similar to a study on oxygen isotope analysis of individual foraminifera from deep marine sediments to document ENSO variability (and annual seasonal cycle) at a few time intervals during the Holocene. I suggest that the authors should focus on the variability rather than mean climate (precipitation). Thanks for the reviewer's comment and suggestion. Your comment affirms that $\delta13C$ of millet can be used as a new proxy to document variability of summer precipitation for the future study, giving us confidence to analyze a larger number of samples per sample at a few time intervals during the Holocene in the future. It is worth noting the variability of summer precipitation as the reviewer's suggestion. The summer precipitation indeed becomes more and more variable especially after ∼5.2 ka BP, which we have added in the discussion of the manuscript. Since 3-5 millets of a sample from the cultural layer probably formed in the interval of several decades rather than 3 to 5 years, the reconstructed results can indicate the mean precipitation as we careful consideration. Although it is hard to conclude the summer monsoon and precipitation decreases during the Holocene, the reconstructed precipitation during the Holocene exhibits the characteristics of a systemic increase with significant fluctuations. We have discussed precipitation fluctuated significantly and captured three markedly humid periods, showing the mean precipitation as well as the increasing variability especially after 5.2 ka BP in the manuscript, hoping the reviewer's agreement.

2. The writing in general is clear – I commend the authors' effort to make it an easy read. However, I find there are many superlative words to describe the results, and some of these are overstatement. I will provide examples below in my specific comments. Specific comments: Title: -focus on precipitation variability? -change "China" to "northern China", for international readership? Thanks for the reviewer's commendation and suggestion. -We have changed "China" to "northern China" following the suggestion. We also consider carefully the precipitation variability in the manuscript but didn't show it in the title.

Abstract -add latitude (34.5 N) and perhaps rounded longitude as well, for international readership? Thanks for the reviewer's suggestion. We have added rounded latitude and longitude on Guanzhong Basin in the abstract.

-there are many superlative descriptors here in the abstract, such as "accurate" (line 17), "robust" (line 19), "reliably" (line 26), "precise" (line 28). It seems to me none of these is needed and justified. Most of similar words should be deleted throughout the text. Thanks for the reviewer's suggestion. To be more justified, we have removed all the mentioned words above in the abstract as well as some similar words throughout the text.

-the abstract needs to refocus if the authors accept my suggestion above. Thanks for the reviewer's suggestion. We partly accepted your suggestion above, that the summer precipitation variability from the early Holocene to late Holocene was increasing, and have refocused the abstract in the manuscript.

Introduction -It is unnecessarily too long. In particular, the general discussion on Holocene climate in the first 3 paragraphs on page 1-2 is not really needed. Delete or shorten. Thanks for the reviewer's suggestion. We have shortened the first 3 paragraphs following the reviewer's suggestion.

-superlative word examples: "accurate" (l 41), "more completely and accurately" (l 60), "robust" (l 135). Thanks for the reviewer's suggestion. We have removed the words referred above from the manuscript.

Line 91: change "between 5.2-4.3 ka BP", to either "between 5.2 AND 4,3 ka BP", or

"at 5.2-4.3 ka BP" (there are other cases of matching "between: : :and: : :" in the text) Methods This section reads well. Thanks for the reviewer's suggestion. We have changed "between 5.2-4.3 ka BP" to "at 5.2-4.3 ka BP" following the suggestion.

Line 156: I wonder if a single seed is large enough for C isotope analysis, but multiple seed analysis still can be used for the variability study as suggested above (but it will be "conservative" reconstruction of precipitation variability, due to averaging of multiple years growth in one sample). Thanks for the reviewer's question. The millet individuals are very tiny and single millet is not enough for C isotope analysis. Therefore, multiple millets were used as a sample for $\delta$13C measurement, which averaging the multiple years growth. So we consider that the reconstruction result can indicate a decadal averaging precipitation. However, the variability can also be concluded from comparison of series of results at different intervals.

L 169: "1r" = 1 sigma? (67% probability?) Thanks for the reviewer's kind remind. Yes, it should be 1 sigma (67% probability) and it's our error writing. So, we have corrected "1r" into "1$\sigma$".

L 179: change to "the sampled culture layers" ("section's" is awkward usage) Thanks for the reviewer's suggestion. We have changed to "the sampled culture layers" following your suggestion.

L 181-186: unclear how it was done. Thanks for the reviewer's expression. To make the readers understand, we have separated these sentences from the preceding as another section entitled "processing data" and added a figure (Figure 3a) with all $\delta$13C and calibrated age range versus depth as well as the groups we built, hoping it helps the reader to understand.

L 189: delete "ref." Thanks for the reviewer's suggestion. We have deleted "ref" following your suggestion.

L 190: delete "," before "demonstrated" Thanks for the reviewer's suggestion. We have

deleted "," before "demonstrated" as you suggestion.

Results L 209: change "eliminating" to "without considering" Thanks for the reviewer's suggestion. We have changed "eliminating" to "without considering" following the suggestion.

Line 207-214: I'm confused here. You describe carbonized Neolithic seed remains and modern common millet, but you compare "modern seeds" in the last sentence. Also, millet is more negative than seeds, rather than "positive" as described. Check. Thanks for the reviewer's kind remind. It should be millets remains and modern millet in the paragraph. To avoid confusion, we have rewrite the sentence as follow: Common millet remains sampled from cultural layers of Guanzhong Basin in our study......It can thus be seen that the $\delta$13C values of common millet remains are more positive than those of modern millet by ï¡đ2.9‰

L 233-235: "slightly higher" and "a much more humid" is contradictory. Overstatement/ over-interpretation? Thanks for the reviewer's kind remind. To avoid contradictory, we have removed "slightly" and "much" from the sentence and the sentence is now as follow: The $\delta$13C values yielded by ancient common millets are higher than those of modern common millet seeds, suggesting that these ancient plants grew in a more humid environment than today's.

Discussion As I commented above, the mean precipitation reconstruction doesn't allow for much comparison and discussion on summer monsoon, while precipitation variability is potentially a novel aspect of paleoclimate research. Although your current data are not robust enough, it seems to me that it holds great promise for the future project: even just 3 or 4 horizons, with large analysis per horizon. Thanks for the reviewer's suggestion. As we answered above, we consider that $\delta$13C of millet can reconstruct mean precipitation although the resolution of current data is low. However, we also accept the reviewer' viewpoint that variability becomes higher from the early Holocene to late Holocene. So we weakened the discussion on the absolute value of the mean

precipitation reconstruction and added discussion about the increasing variability of summer precipitation as well as provided the markedly humid periods.

L 341: change "1961-2011" to "1951-2011"? which makes 60 years and also is consistent from description earlier. Thanks for the reviewer's kind remind. It should be 1951 and we have corrected it to "1951-2011".

Conclusions L 392-393: "low resolution" and "convincing" are contradictory. Thanks for the reviewer's kind remind. To avoid contradictory, we have changed "convincing" to "innovative".

Tables Table 1 -move latitudes and longitudes from Figure 1 to new columns here. -what "source" means here? "12" here means "12 culture layers"? if so, spell out. -change "No." to lower case and italic "n" (to indicate number of analyses or samples) Thanks for the reviewer's suggestions. -We have added latitudes and longitudes in a new column. -The "sources" means "sample source" and we have added "sample" before "sources". "12" here was placed in an error box and we have moved it to the right box. -We replaced "No." by "n" and gave a footnote "n means the number of remnant common millet samples derived from the section".

Table 2 -Change heading to " AMS 14C dates" ("dating data" is unusual) -why does a 250-cm-long section (such as BN) have the same or reversed ages? Very rapid accumulation of these layers? I hope it is discussed elsewhere in archeological literature. -change the heading of column 4 to "AMS 14C date (yr BP)" – it is wrong to say 14C date as "cal yr BP" -maybe a footnote to indicate the dating lab for OZM Thanks for the reviewer's suggestions. -We have changed heading to "Accelerator mass spectrometry (AMS) dates from Baijia (BJ), Huiduipo (HDP), Manan (MN), Beiniu (BN), and Nansha (NS)". -Since all sections selected were cultural layers and deeply affected by human activities, the sections especially from the ash pits accumulate rapidly, which may result in the same ages, and have reverse layer due to disturbance of human activities. The thickness of the sections and the cultural types were discussed in the archeological

literature "Atlas of Chinese Cultural Relics: Shannxi Municipality", which was edited by State Cultural Relics Bureau in 1998, in Chinese. However, the 14C dates was not adopt in the literature, due to archaeologists usually hold the view that archeological periodization are more reliable. Our radiocarbon dates showed the cultural layers are different from the natural layer. That is why we cannot apply linear interpolation and extension for age verse depth as which usually applied to the natural section. -We have changed the heading of column 4 to "Radiocarbon age (14C yr BP)". -We have added a footnote for the dating lab of OZM in the manuscript as follow: All assays were run on the STAR Accelerator, ANSTO, Australia.

Table 3 -maybe indicate ages as "calibrated ages" to avoid confusion -add footnote to indicate "d13Cre" and "Ps" Thanks for the reviewer's suggestions. -We have changed "14C age" to "calibrated ages" following your suggestion. -We have changed "$\delta$13Cre" to "corrected $\delta$13C" and also added footnote to indicate corrected $\delta$13C and Ps.

Figure 1. -move latitude and longitude to Table 1 -CorelDRAW12 is not needed to mention, as it is just a map. Thanks for the reviewer's kind remind. We have moved latitude and longitude to table 1 and removed the sentence mentioned CorelDRAW12 from the footnote.

Figure 2. -need more explanation about panel a (raw data points) and b (box plot) in figure caption -again Fig. 2a kind of shows increase in precipitation variability from 8 ka to 3 ka. Have you tried a regression of all the data to see if there is a significant decline (in precipitation) during that period as well? (perhaps the number of data points are still low) Thanks for the reviewer's suggestion. -We have added more explanation about panel a and panel b in figure caption in the manuscript following your suggestion. -To be more specific, we redrew the figure 2a, showing all raw data points including $\delta$13C and calibrated age range versus depth. However, a significant decline in $\delta$13C or precipitation from 8 ka BP to 3 ka BP wasn't concluded. But the increasing variability of precipitation is visible, which we have discussed in the manuscript.

Figure 3 -indicate reference in figure caption "Modified or data from Yang and Li (2015)" Thanks for the reviewer's suggestion. We have added "which data from Yang and Li (2015)" in the figure caption.

Figure 4 -Again, I don't think it is a good way to present the data as groups to get mean climate/precipitation – considering the large variability almost nothing can be concluded here (a.k.a. the pattern is not robust/convincing, because of uncertainty). Thanks for the reviewer's suggestion. To better present, we have added another panel as figure 5a to display all reconstructed precipitation data, hoping it is helpful to understand the mean precipitation and variability. It can be seen from the panel a that there are three markedly humid periods, which have the mean precipitation higher than the other periods, and the variability of precipitation from 8 ka BP to 3 ka BP becomes increasing obviously.

Please also note the supplement to this comment:
http://www.clim-past-discuss.net/cp-2016-87/cp-2016-87-AC2-supplement.pdf

---

## Author Response (AR1)

Dear editor:

   We would like to express our feelings of appreciations to you for your kindly help and professional comments to our manuscript entitled "Summer precipitation reconstructed quantitatively using a Mid Holocene $\delta^{13}$C common millet record from Guanzhong Basin, northern China". We have tried our best to modify the weakness and flaws pointing out by you. The revised places were marked by blue font color, on the basis of pre-revised version which was marked by red font color. Now, we believe that we made a better work which would probably satisfy the editor and suitable to be published. The answer to the comments is listing in the following paragraph.

      Thanks again for your help.
      Best wishes!

Sincerely,

Yang Qing and Xiaoqiang Li

**Editor's comments:**

   Thank you for submitting your work to CPD. As you have seen, both referees considered that d13C of millet seed is a suitable proxy for quantitative construction of summer precipitation in northern China, and your study is worthy a publication in Climate of the Past after revisions. Towards this end, they both suggested a number of issues that you need to consider for improving the manuscript.

   I have read carefully your responses and your pre-revised version, and found that you had considered most of their suggestions. However, I estimate that a further revision is needed on the following issues.

1. Both referees indicated that the text contains many superlative words or descriptions. Although your pre-revised version has toned down, there are still some lefts. As an example, lines 28 & 395: 'innovative proxy'. To my view, 'new proxy' is largely enough. I request you to carry out a thorough check on this kind of problems.

   Thanks for the editor's suggestion. We have further toned down according to the suggestion. Details are as follows:
      Line 15: changed 'explicit' into 'suitable'.
      Line 15: changed 'accurate' into 'faithful'.
      Line 28: changed 'innovative proxy' into 'new proxy'.
      Line 87: changed 'unambiguous' into 'clear'.
      Line 104: changed 'perfect' into 'preferable'.
      Line 385: changed 'innovative proxy' into 'new proxy'.

2. As also indicated by the referees, the writing of the ms is in overall understandable, but is not yet in good English. Although you already made some improvements in the pre-revised version, a thorough language revision by a native English speaker is still indispensable for

publishing in CP. For examples, it would be finer to revise the title as "Summer precipitation reconstructed quantitatively using…"to "Quantitative reconstruction of summer precipitation using…". Line 14: the expression "to produce quantitative Holocene precipitation reconstructions…" is also not satisfactory. There are still many problems of this kind that I am not intending to list here. These could be fixed along with the issue of "superlative words'.

Thanks for the editor's suggestions. The manuscript has been revised by a native English speaker according to the suggestion. And to be more satisfied, we have rewritten the title as 'Quantitative reconstruction of summer precipitation using a Mid Holocene $\delta^{13}$C common millet record from Guanzhong Basin, northern China'. We also revised Line 14 as 'To quantitatively reconstruct Holocene precipitation for particular geographical areas…', hoping it's more understandable and straightforward.

3.  Given that the ms is not long, I think it would be better to remove the subtitles 3.1…, 5.1… in view of the serious length imbalance of them, and in view of some inaccurate expressions (e.g. 5.2, 'validating the reliability of quantitative precipitation reconstruction'). You didn't really 'validated' it, but just argued/discussed it.

Thanks for the editor's kindly suggestions. We have removed the subtitles 3.1, 3.2…, 5.1…according to your suggestion.

4.  I am not satisfied with the 'conclusion' section. It currently only focuses on repeating the estimated results but the significance of this new proxy is not really concluded. I would think the estimated results in this study are rather tentative, but the study provides a useful method for estimating the paleoclimate conditions of specific archeological layers that usually interests the archaeology community. It might be better to mention this.

Thanks for the editor's kindly suggestions. We have refocused the 'conclusion' section according to the editor's suggestion, removing some repeated sentences and adding the significance of the new proxy in to the section, as following:

'$\delta^{13}$C of common millet from archaeological layer can effectively record precipitation during millet growing season…. The work provides a new proxy for establishing the paleoprecipitation record.

Charred common millet remains continuously exist in the archaeological layers since around 10 ka in northern China. Not only the common millet can provide a reliable dating framework, but also the continuous $\delta^{13}$C-based paleoprecipitation sequences could be quantitatively reconstructed….'

5.  The 'summer' here refers to the interval from mid-June to September, please also specified in the abstract.

Thanks for the editor's kindly suggestions. We have added the specific interval of summer in the abstract according to the suggestion, as in lines 19-20.